# An Effective Primary Head and Neck Squamous Cell Carcinoma In Vitro Model

**DOI:** 10.3390/cells8060555

**Published:** 2019-06-07

**Authors:** Felix Oppel, Senyao Shao, Matthias Schürmann, Peter Goon, Andreas E. Albers, Holger Sudhoff

**Affiliations:** 1Department of Otolaryngology, Head and Neck Surgery, Klinikum Bielefeld, Teutoburger Str. 50, 33604 Bielefeld, Germany; felix.oppel@klinikumbielefeld.de (F.O.); docssy0527@gmail.com (S.S.); matthias.schuermann@klinikumbielefeld.de (M.S.); 2Department of Dermatology, Addenbrooke’s Hospital, Hills Road, Cambridge CB2 2QQ, UK; peter.goon@nhs.net; 3Department of Otolaryngology, Head and Neck Surgery, Charité-Universitätsmedizin Berlin, 10117 Berlin, Germany; andreas.albers@charite.de

**Keywords:** head and neck cancer, HPV, primary cell culture, stroma, tumor spheroids

## Abstract

Head and neck squamous cell carcinoma is a highly malignant disease and research is needed to find new therapeutic approaches. Faithful experimental models are required for this purpose. Here, we describe the specific cell culture conditions enabling the efficient establishment of primary cell culture models. Whereas a classical 10% serum-containing medium resulted in the growth of fibroblast-like cells that outcompeted epithelial cells, we found that the use of specific culture conditions enabled the growth of epithelial tumor cells from HPV+ and HPV− head and neck cancer tissue applicable for research. EpCAM and high Thy-1 positivity on the cell surface were mutually exclusive and distinguished epithelial and fibroblast-like subpopulations in all primary cultures examined and thus can be used to monitor stromal contamination and epithelial cell content. Interestingly, cells of an individual patient developed tumor spheroids in suspension without the use of ultra-low attachment plates, whereas all other samples exclusively formed adherent cell layers. Spheroid cells were highly positive for ALDH1A1 and hence displayed a phenotype reminiscent of tumor stem cells. Altogether, we present a system to establish valuable primary cell culture models from head and neck cancer tissue at high efficiency that might be applicable in other tumor entities as well.

## 1. Introduction

Head and neck cancer is a highly malignant disease with 600,000 new incidences worldwide per year and a mortality rate of about 50% [1,2]. About 90% of all tumors in the head and neck area are head and neck squamous cell carcinomas (HNSCCs) and occur in heterogeneous locations. They arise from the mucosal epithelium of the oral cavity, larynx, and pharynx [3]. HNSCCs in oropharyngeal location can be subdivided into cases caused by the human papilloma virus (HPV) and those cases that are HPV-negative, which differ strikingly in their molecular biology [4]. The *E6* and *E7* oncogenes of HPV inactivate p53 and pRB, causing carcinogenesis [5], whereas HPV-negative tumors show genetic alterations in tumor suppressor genes like *TP53* and *CDKN2A* and oncogenes like *EGFR* and *PIK3CA* [6]. HPV-negative oropharyngeal HNSCCs have a worse prognosis than HPV-positive tumors [7], and the survival of HPV-negative HNSCC patients has not improved substantially in recent decades [8], so new therapeutic approaches are needed to treat this malignancy.

Primary tumor cell cultures are important tools in cancer research as they resemble the characteristics of individual patients’ tumors much closer than decade-old permanent cell lines. In order to search for vulnerabilities in cancer cells, primary cell culture models reflecting individual patients’ tumors provide high potential for investigating new therapy approaches and personalized medicine [9]. However, the establishment of primary cancer cell cultures from patient-derived tissue can be challenging due to insufficient tumor cell survival and benign contaminations. For HNSCC research, primary cell lines were established in previous studies, either from single cells derived from enzymatically dissociated tumor material [10,11], explant cultures [12], or a combination of both [13]. To expand primary cells in culture MEM, DMEM, RPMI-1640, and DMEM-F12 medium containing 5–20% fetal bovine serum (FBS) [10,11,12,13,14,15,16,17], or serum-free DMEM-F12 supplemented with epidermal growth factor (EGF) and basic fibroblast growth factor (bFGF) [14,15,17] has been utilized. Also, feeder layers consisting of growth-impaired fibroblasts have been used to support primary HNSCC cell expansion in vitro (reviewed in [18]). 

Serum-free culture conditions are generally believed to support the growth of more undifferentiated stem-like tumor cells, reminiscent of so-called cancer stem cells (CSCs) [19]. These comprise a subpopulation of cells within a tumor capable of self-renewal, supporting long-term tumor growth, and are frequently hypothesized to have the unique capability to grow anchorage-independent as suspension spheroids in serum-free media. The enrichment of this kind of cells in a primary cell culture might lead to a cell culture model that can serve as a basis for the establishment of targeted strategies eradicating the stem cell root of tumor diseases. Thus, in HNSCC research spheroids from primary tumor cells and permanent cell lines have been used as model systems in previous studies [14,15,17,18,20,21]. Using these spheroids, CSC populations in HNSCC have been identified. Expression of Aldehyde Dehydrogenase 1 Family Member A1 (ALDH1A1) has been identified to mark a subpopulation of HNSCC cells with increased tumorigenic potential in xenotransplantation assays in immunodeficient mice [22,23,24]. In human HNSCCs, ALDH1A1 expression correlates with lower tumor differentiation and worse prognosis [25,26]. In general, ALDH1A1 is a known maker of stem cells in normal tissues and various tumor types and regulates cellular processes like self-renewal, proliferation, and repression of apoptosis (reviewed in [27]). 

However, primary cell cultures are frequently contaminated with cancer-associated fibroblasts (CAFs). As described previously in pancreatic cancer, fibroblast-like cell types from the tumor-associated stroma were found to survive, proliferate, and contaminate primary cell cultures even under serum-free culture conditions [28,29]. Similar findings resulted from HNSCC primary cultures and the removal of contaminating CAFs was attempted by serial trypsinization, where more loosely attached CAFs detach earlier compared to the epithelial tumor cells, and/or by cell scraping [11,12,15,18], partially with limited success [11]. Thus, benign contaminations in primary cultures can potentially bias cancer research studies if unidentified or neglected. A common strategy to eradicate human CAFs is the transplantation of patient-derived tumor tissue into immunodeficient mice to allow xenograft-tumor formation, since CAFs cannot expand permanently in vivo [28]. Xenograft-tumor models have previously been established from HNSCCs and are highly suitable for the expansion of primary HNSCC tissue and to evaluate new therapeutic approaches [30,31]. However, animal tumor models are more expensive, hence smaller in scale, and ethically more controversial than in vitro cultures. Moreover, not all laboratories can afford or facilitate animal experiments. Thus, protocols for the more efficient establishment of cell culture models would benefit HNSCC research. 

In this study, we describe a primary HNSCC cell culture model that allows for the effective establishment of primary HNSCC cultures using simple techniques without the need for xenotransplantation, feeder layers, serial trypsinization, or scraping of contaminating CAFs. Moreover, we examined the dynamics of the proliferation of tumor and stroma cells in different growth media and found suitable markers to identify the extent of CAF contamination of primary tumor cell cultures. Our primary cell cultures can be used in future studies to investigate tumor cell biology, personalized medicine approaches, and to conduct pharmacology screens. 

## 2. Materials and Methods

### 2.1. Human Materials

Primary head and neck cancer tissue was obtained from medically indicated surgeries with informed consent of the patients, according to the declaration of Helsinki, and as approved by the ethics committee of the Ruhr-University Bochum (AZ 2018-397). 

### 2.2. Tumor Purification

Tumor tissue was minced into approximately 2-mm pieces and washed twice with PBS. Pieces were vigorously shaken in PBS before each centrifugation step to enhance the washing effect and reduce the probability of contamination. Pieces were digested using 5–10 mL of a 2.5 mg/mL Collagenase NB4 standard (Nordmark Biochemicals, Uetersen, Germany) solution in PBS + 3mM CaCl_2_ for 1.5–2 h at 37 °C with gentle vortexing every 5–10 min. After that, the digestion mix was diluted with 30 mL PBS and pipetted up and down to further detach aggregates. Single cells and partially digested tumor pieces were separated by filtration and placed in separate culture dishes. Whenever larger tumor pieces were left undigested by the procedure, these were filtered out using a 100-µm cell strainer (BD Biosciences, Heidelberg, Germany) and cultured separately. Alternatively, filtration was excluded and a mixture of single cells, aggregates, and tumor pieces was taken into culture. Erythrocytes were lysed using a sterile erythrocyte lysis buffer (containing 0.802 mg/mL ammonium chloride, 0.084 mg/mL sodium bicarbonate and 0.037 mg/mL disodium EDTA in H_2_Odd) for 10 min at 37 °C. After digestion, purified cells/cell aggregates were washed twice in PBS and re-suspended in culture medium. 

### 2.3. Cell Culture

Purified tumor cells and pieces were monitored until a 60–80% confluent adherent cell layer was formed. These fresh cultures were defined as passage 0. Adherent cells were subsequently washed with PBS and detached using Accutase (Capricorn Scientific, Ebsdorfergrund, Germany) and split 1:2 to 1:6. To split spheroids, these were washed in PBS and disintegrated in a 37 °C water bath using Accutase with gentle vortexing every 5–10 min. After that, cells were washed in PBS, then re-suspended in culture medium and incubated at 37 °C with 5% CO_2_. Cell numbers were calculated using a Neubauer Chamber. To monitor exponential growth the counted cell number was multiplied by the split rate. Imaging of cell cultures was performed using an Olympus CKX41 microscope (Olympus Deutschland GmbH, Hamburg, Germany). Culture medium compositions: DMEM or RPMI-1640-medium (Capricorn Scientific) was supplemented with 10% fetal bovine serum (FBS) (Sigma-Aldrich, St. Louis, MO, USA), 1% 200mM L-Glutamine, 1% 100x Penicillin/Streptomycin, and 1% 250 µg/mL Amphotericin B solution (all three Capricorn Scientific). CSC-medium was composed of 500 mL DMEM-F12 Ham medium (Sigma-Aldrich), supplemented with 1% 200 mM L-Glutamine, 1% 100x Penicillin/Streptomycin, 1% 250 µg/mL Amphotericin B solution, 0.6% glucose, 12 μg/mL heparin (Sigma-Aldrich), 10 mL NCS21 Supplement (50x) (Capricorn Scientific), 10 ng/mL FGF basic (PeproTech, Hamburg, Germany) added twice a week, and 20 ng/mL EGF (PeproTec) added twice a week. PNEU-medium was based on PneumaCult™-Ex Plus Medium including 10 mL of the 50x PneumaCult™-Ex Plus supplement (STEMCELL Technologies Inc., Vancouver, Canada), supplemented with 1% 200 mM L-Glutamine, 1% 100x Penicillin/Streptomycin, 1% 250 µg/mL Amphotericin B solution, and 0.5 mL of a 96 µg/mL (0.2 mM) hydrocortisone solution. 

### 2.4. Indirect Immunofluorescence

Indirect immunofluorescence was performed as described previously [32]. Spheroids in suspension were stained applying the same protocol but using centrifugation between every step. Primary antibodies: rabbit-anti-human EpCAM (EGP40/1556R; 1:100; Novus Biologicals, Centennial, CO, United States), rabbit-anti-human cytokeratin 14 (1:200, Thermo Fisher Scientific, Waltham, MA, United States), rabbit-anti-human cytokeratin 19 (1:75; Novus Biologicals), mouse-anti-human Vimentin (V9; 1:100; Santa Cruz Biotechnology Inc., Dallas, TX, United States), mouse-anti-human α-smooth muscle actin (1A4; 1:200; Sigma-Aldrich), mouse-anti-human Thy-1 (AS02; 1:100; Dianova, Hamburg, Germany), rabbit-anti-human ALDH1A1 (20H2L4, 1:100; Thermo Fisher Scientific), and rabbit-anti-human CDKN2A/p16INK4a (EPR1473; 1:200; Abcam, Cambridge, UK). Secondary antibodies were goat-anti-mouse-IgG-Alexa Fluor 555 and donkey-anti-rabbit-IgG-Alexa Fluor 488 (both Thermo Fisher Scientific). DNA was stained with Hoechst 33,342 and the actin cytoskeleton was visualized using phalliodin-PF647 (Promokine, Heidelberg, Germany). Imaging was conducted using a confocal laser scanning microscope CLSM 780 (Carl Zeiss, Oberkochen, Germany) and ZEN software (Carl Zeiss).

### 2.5. Quantitative Real-Time Polymerase Chain Reaction (qRT-PCR)

The RNA isolation was achieved by the aid of the innuPREP RNA Mini Kit (Analytic Jena AG), according to the manufacturer’s guidelines. The quality and quantity of the eluted RNA were measured by a nanophotometer (Implen GmbH, Munich, Germany). The cDNA synthesis was executed with random hexamer primers by the RevertAid First Strand cDNA Synthesis Kit (Thermo Fisher Scientific, Inc.), applying the standard protocol from the user’s guide. 

Depending on the initial RNA concentration, the obtained cDNA solution was diluted in water to a ratio between 1:5 and 1:30. To detect very rare products, the initial cDNA solution was slightly diluted, with a 1:1 ratio. For all qPCR reactions, Luna^®^ Universal qPCR Master Mix (New England Biolabs, Ipswich, MA, United States) was employed. According to the manufactures guidelines, the final concentration of utilized primers was set to 0.25 µM. All reactions were executed as technical triplicates. For product detection, the MIC qPCR cycler (Bio Molecular Systems, Upper Coomera, QLD, Australia) was employed, running the recommended cycling protocol. Glyceraldehyde 3-phosphate dehydrogenase (GAPDH) expression was utilized for relativization of cycle threshold values. GraphPad Prism (GraphPad Software, San Diego, CA, United States) was used for graphics and statistical analysis. The highest expression value in the analysis was normalized to 100%.

Primer sequences: GAPDH-for CTGCACCACCAACTGCTTAG, GAPDH-rev GTCTTCTGGGTGGCAGTGAT; EpCAM-for: TAAGGCCAAGCAGTGCAACG, EpCAM-rev: TTGTCTGTTCTTCTGACCCCAG; Thy-1-for: CAGCAGTTCACCCATCCAGT; Thy-1-rev: TGGTGAAGTTGGTTCGGGAG; CK14-for: CCTCCTCCAGCCGCCAAATCC, CK14-rev: TTGGTGCGAAGGACCTGCTCG; CK19-for: GAATCGCAGCTTCTGAGACCA, CK19-rev: CTGGCGATAGCTGTAGGAAGT; ALDH1A1-for: TTGTTCCTGGTTATGGGCCT, ALDH1A1-rev: GCTGGCAATGCAGACATTCTTA; Vimentin-for: CAGGACTCGGTGGACTTCTC, Vimentin-rev: GAAGCGGTCATTCAGCTCCT; α-SMA-for: GTTCCGCTCCTCTCTCCAAC, α-SMA-rev: ACGCTGGAGGACTTGCTTTT.

## 3. Results

### 3.1. Fibroblast-Like Cells Overgrow Epithelial Tumor Cells in Primary HNSCC Cell Cultures under Standard Conditions

In order to establish primary cell cultures, head and neck cancer tissue samples were taken during medically required surgeries from 25 patients, of whom 22 were diagnosed for squamous cell carcinomas (Appendix A). Both, p16-positive and p16-negative samples were included. The tissue was dissected and digested enzymatically. Initially, DMEM or RPMI-1640 medium supplemented with 10% FBS was used, according to previous studies on primary HNSCC cultures. Within four weeks, cells and tumor pieces attached to the culture dish. In eight out of nine cases (88.9%) (Samples S1–S9), spindle-shaped mesenchymal cells, reminiscent of fibroblasts, were observed (Appendix A). In three cases (33.3%), we additionally detected adherent colonies of epithelial cells surrounded by mesenchymal cells (Figure 1A). These initial cell cultures (passage 0) were enzymatically detached and passaged up to 10 times. However, within the first 1–3 passages epithelial cells vanished and a monolayer of fibroblast-like cells formed in all cases. During this process we observed phenotypic changes in the epithelial cell compartment (Figure 1B). In particular, cells of S3, S4, and S5 displayed scattered patches of a detaching dead cell layer, appearing as bubble-like structures or as thick contracted cell spindles. Dead or dying cells displayed more prominent intracellular structures, indicating a cross-linked cytoskeleton and white inclusions resembling lamellar bodies reminiscent of cell death by cornification. Analysis of EpCAM as a general epithelial marker, cytokeratin 14 and 19 (CK14 and CK19) as epithelial markers of HNSCC cells [33], and Thy-1, α-smooth muscle actin (α-SMA), and Vimentin as markers of stromal CAFs was subsequently performed by Q-PCR using culture cells from patient sample 4 (S4), which initially showed the largest epithelial cell content. EpCAM, CK14, and CK19 were significantly downregulated between passage 1 and passage 2 (P1 and P2), whereas all three stromal markers were strongly upregulated (Figure 1C). This demonstrates a loss of the epithelial compartment in DMEM-grown primary HNSCC cell cultures, accompanied by an outgrowth of CAFs. 

### 3.2. Distinct Serum-Free Media Differ in Their Ability to Support the Growth of Primary Epithelial HNSCC Cells 

In order to better facilitate the expansion of epithelial tumor cells, we tested two distinct serum-free media. The first was DMEM-F12 medium supplemented with EGF and bFGF, adapted from previous studies [28,34], and further referred as cancer stem cell medium (CSC-medium). The second was Pneumacult Ex Plus basal medium, which was originally designed to support growth of airway epithelial cells, and hereafter will be referred as PNEU-medium. Out of six tumor samples (S12-S15, S17, and S18) that were purified and initially cultured in CSC-medium, four showed survival of epithelial tumor cells (66.7%; S12, S15, S17, and S18, Appendix A). However in two of these four cases, epithelial cells only attached to the culture dish without detectable expansion (S17 and S18). Only two samples (33.3%; S12 and S15) showed striking proliferation in CSC-medium in P0 and were initially passaged after eight (S12) and 17 days (S15). In contrast, five out of seven samples (71.4%) successfully attached to the culture dish and formed expandable epithelial cell cultures in PNEU-medium. To study the differences in the proliferation and marker expression, S12 and S15 cells were cultured in parallel in 10% FBS-containing DMEM, serum-free CSC-medium, and serum-free PNEU-medium. 

Serum-free cultures of both patients initially showed adherent growth of epithelial tumor cells in passage 0 (Figure 2A). However for S12, we observed strong proliferation until passage 1, which stopped in passage 2 for all three different media (Figure 2B). For S15, a strong increase in cell number was measured until passage 2 in PNEU and DMEM-medium, whereas cells showed poor expansion in CSC-medium. In passage 3 the overall cell numbers decreased (Figure 2B), except in DMEM. However in DMEM, S15-derived epithelial cells frequently acquired the above-described cornification-like phenotype and were overgrown by cells displaying CAF-like morphology (Figure 2C). As PNEU-medium appeared to be more suitable for the expansion of primary epithelial HNSCC cells than CSC-medium, all further serum-free cell culture experiments were performed in PNEU-medium. In contrast to S12 and S15, cells of S18 behaved strikingly different in culture. In DMEM-medium containing 10% FBS, no expansion of any cell type of S18 was observed. Only individual fibroblast-like cells attached but showed no signs of proliferation (Appendix A). In PNEU-medium, epithelial cells grew as an adherent monolayer (S18-AD) until passage 3 (Figure 3A). Besides the adherent monolayer, spheroids formed in the supernatant (S18-SPH) without the use of ultra-low attachment plates. From passage 4 on, the adherent S18 cells did not re-attach and also turned into a spheroid culture, which was visually indistinguishable from the spheroids that grew since passage 0. Cells derived from S22, S23, S24, and S25 grew as a monolayer of epithelial cells and not as spheroids, despite being processed in the same way as S18 (Figure 3A). S18-AD and S18-SPH cultures in PNEU-medium grew exponentially (Figure 3B). Cells of S22 stopped growing in passage 6 and displayed expansion of fibroblast-like cells. Cells of S23 displayed the same phenotype in passage 1 and were thus not cultured further. In contrast to that, cells derived from S24 grew exponentially for the first six passages. S25 cells proliferated continuously but were not followed beyond passage 3, as S25 was derived from a squamous cell carcinoma of the outer ear skin and thus represented a different tumor entity than S12, S15, S22, S23, and S24, which were established from mucosal HNSCCs (Appendix A). Interestingly, tumor samples S12, S15, and S24 were identified to be p16-positive in histopathology (Appendix A), which in clinical practice serves as a surrogate marker for HPV infection. 

When S22 and S24 cells were split from PNEU-medium into DMEM containing 10% FBS, epithelial cells disappeared and fibroblast-like cell types expanded and replaced the epithelial population upon further passaging (Figure 4A). This was accompanied by a loss of EpCAM expression and a strong increase in Thy-1 over six passages (Figure 4B). This shows that even in established cell cultures residual stromal cells were able to overgrow epithelial cell populations in classical serum-containing culture conditions. 

Thus, our system is suitable to isolate cell cultures from both, HPV-positive and HPV-negative HNSCCs. Frozen cells of S4 that originally showed epithelial cell failure in DMEM-medium, as described above, were thawed and cultured in PNEU-medium. Strikingly, epithelial cells proliferated and overgrew fibroblast-like cell types within two further passages (Appendix A), demonstrating the potential of PNEU-medium to support the growth of epithelial HNSCC cells in vitro. 

### 3.3. ALDH1A1 Expression Varies among Distinct Patients’ HNSCC Cell Cultures

The purpose of our cancer model is to facilitate the expansion of disease-relevant tumor cells, we tested for expression of ALDH1A1, a described marker of cancer stem cells associated with inferior survival in HNSCCs [26,35]. Because ALDH1A1 is an intracellular protein, qRT-PCR was used to compare expression levels between the original tumor tissue and within the first three passages of cell cultures derived from patient samples S12, S15, S18, S22, and S24. 

When the expression values were normalized by each patient individually, we observed much higher ALDH1A1 expression for samples S12, S22, and S24 in the original tumor than in cell culture (Figure 5A). For S12, S22, and S24, expression gradually decreased over the passages. For S15, ALDH1A1 expression was comparable in culture and in the original tumor. In contrast, for adherent S18 cell cultures, ALDH1A1 expression gradually increased until passage 3 by about 42 times. 

When the same dataset was normalized by passage, tumor S24 was found to have the highest ALDH1A1 expression compared to all other tumors (Figure 5B). However from passage 0 on, S18 cells displayed much higher expression than any other culture. When the dataset was finally normalized to compare all samples of all passages and patients, we observed strikingly higher ALDH1A1 levels in S18 passage 3 than in any other sample (Figure 5C). Thus, the dynamics of ALDH1A1 expression during cell culture establishment varied strongly between different patients’ samples. For individual samples, the use of PNEU-medium can select cells with high ALDH1A1 expression, reminiscent of previously described stem-like tumor cell populations [35].

### 3.4. The Composition of HNSCC Cultures Can Be Accessed by Differential Marker Expression 

In order to serve as a high-value model system for cancer research, patient-derived cell cultures must exhibit a degree of purity that is dependent on the specific application. Thus, we examined expression of the above described epithelial markers EpCAM, CK14, and CK19, as well as mesenchymal markers Thy-1, Vimentin, and α-SMA for their suitability to indicate the expansion of the epithelial compartment and the extend of stromal contamination. Expression levels were examined by qRT-PCR and indirect immunofluorescence under different culture conditions and associated with cell morphology. 

As described above, S12 and S15 cell cultures were established in CSC-medium and then passaged in parallel in the three different media DMEM, CSC, and PNEU until they stopped proliferating in passage two or three. In DMEM, epithelial markers were much higher expressed in the original tumor tissue than in cell culture (Appendix A). However, this was not observed for the mesenchymal markers, which mainly increased simultaneously in vitro whenever DMEM was used. EpCAM and CK19 were constantly expressed in serum-free PNEU and CSC media, whereas expression values decreased to low or zero in DMEM (Appendix A). Altogether, this correlated well with the loss of epithelial cells accompanied by stromal CAF proliferation in DMEM described above for S3, S4, and S5 and the improved expansion of the epithelial compartment in PNEU. Despite epithelial expansion in PNEU-medium, a subpopulation of cells displaying CAF morphology was occasionally observed to surround epithelial colonies that were found to be negative for EpCAM but stained highly positive for cell surface Thy-1 (Figure 6). All epithelial EpCAM+ cells were negative for Thy-1, except for S18 spheroid cells, displaying weak Thy-1 cell membrane localization (Figure 6). Cells co-expressing Thy-1 and EpCAM were found to be bi- or multinuclear reminiscent of fusion cells (Appendix A). CK14 marked a subpopulation of epithelial cells that varied strongly between different patients (Appendix A). CK19 was not detected in mononuclear cells with stromal morphology, but the vast majority of epithelial cells stained positive for this marker (Appendix A). Vimentin marked all cells with fibroblast morphology and was heterogeneously expressed in epithelial cells (Appendix A). However, α-SMA was heterogeneously expressed in both the epithelial and the mesenchymal compartment (Appendix A). 

Thus, EpCAM and CK19 identify epithelial cells in our primary HNSCC cultures, whereas high Thy-1 indicates stromal CAF contaminations (Table 1). CK14, Vimentin, and α-SMA were not suitable for distinction in our study.

## 4. Discussion

Primary cell cultures derived from surgically excised human tumor tissue are highly valuable research tools to investigate the biology and therapeutic vulnerability of cancer cells considering the genetics of each individual neoplasm. Here, we describe a suitable medium (PNEU, see below for composition) that is commercially available and enables the expansion of epithelial cells from patients’ tumors with a high success rate of more than two-thirds. Traditionally, researchers used media like MEM, DMEM, or RPMI-1640 containing serum to expand primary cancer cells [10,11,12,13,14,15,16,17]. However, in our study CAF-like cells outcompeted epithelial tumor cells in serum-containing medium and our data show that it compromises the proliferation and survival of epithelial tumor cells. This shows that classical DMEM + 10% FBS is neither suitable to expand HNSCC cells with epithelial morphology in vitro nor to preserve phenotypically undifferentiated cell populations within cultures.

In contrast, serum-free CSC-medium facilitated epithelial cell proliferation at least for a subset of patients, which is in line with previous publications using this medium in similar compositions for primary cell culture of HNSCCs and other malignancies [10,11,12,13,14,15,16,17,28,29,34]. However both, overall cell culture establishment efficiency and the proliferation rate of epithelial tumor cells, were higher in PNEU-medium. Moreover, spheroids of S18 containing clonally expandable self-renewing cells were only observed in PNEU-medium, but not in CSC-medium. Thus, PNEU-medium can support the self-renewal of individual HNSCC cell clones without the need for prior expensive and difficult enrichment in xenograft-tumor models, which was shown to be necessary in other tumor types, e.g., in pancreatic cancer [28,29]. PNEU-medium was originally designed to support the expansion of primary human airway epithelial cells; we discovered its suitability for HNSCC cell culture by testing this promising candidate. It does not seem to kill stromal fibroblasts but provides a selection advantage to epithelial tumor cells. Thus, PNEU-medium might also be interesting for the creation of in vitro models of other cancer types, e.g., lung cancer. The reason why primary HNSCC cells require specific culture medium conditions remains unknown. However, PNEU-medium appears to protect epithelial cells and facilitates their proliferation so that they are not overgrown by fibroblasts. Furthermore, specific factors in classical media might trigger the cornification observed here.

Whenever CAF-like cells expanded and epithelial cells failed in primary cultures, this was accompanied by increased Thy-1 expression and decreased EpCAM/CK19 expression. Immunostaining for these markers revealed that Thy-1 and EpCAM or CK19 were never observed on a protein level to co-locate in mononuclear cells, except for S18 spheroid cells, which displayed low levels of Thy-1 compared to CAFs (Figure 5). Thus, we conclude that EpCAM and high Thy-1 can be used as markers to differentiate between epithelial tumor cells and stromal CAFs and to determine the extent of stromal contamination in primary epithelial HNSCC cell cultures, which is important to assess the validity of experimental data derived from assays using primary cell cultures.

However, it is not clear whether the CAFs observed in our cultures are of benign or neoplastic origin, e.g., derived by epithelial-to-mesenchymal transition (EMT), which was previously observed in HNSCC lines [36]. However, rare cells in our cultures, displaying both EpCAM and Thy-1, were multinuclear and thus appeared to be derived from tumor/stroma cell fusions rather than from EMT. Even though epithelial markers are frequently downregulated in EMT tumor cells, we would expect some cells to display intermediate phenotypes, as reviewed previously [37,38]. We observed cell populations co-expressing EpCAM and CK19 with combinations of Vimentin and α-SMA, but never strong levels of Thy-1. This indicates that cells with high Thy-1 expression might represent a non-neoplastic compartment within primary HNSCC cultures. Minor contamination with these cells appears to be unavoidable, when cell cultures are established directly from patient-derived tumor tissue. Certainly, benign cells can bias biological assays and hence their abundance relative to epithelial cells should be closely monitored using the marker Thy-1. Our data indicate that primary cell culture establishment should be attempted using distinct media conditions in parallel, whenever the available amount of tumor tissue is sufficient. Primary stromal fibroblasts can be expanded in DMEM, whereas serum-free media can be employed to select epithelial cells. This allows in vitro assays addressing the interaction of both compartments.

Interestingly, S18 spheroids grew in suspension without the use of ultra-low attachment culture dishes. These spheroids were observed in passage 0 of S18 cell, along with adherent monolayer cells that resembled the adherent cells derived from other tumor samples like S22 or S24. More remarkably, these initial adherent S18 monolayer cells transformed over the first four passages into spheroids as well, which appeared like the original spheroids derived from passage 0. This was preceded by an exponential increase in ALDH1A1 expression, indicating that cells of an undifferentiated type were selected in PNEU-medium. To our knowledge, no comparable spheroid model exists for HNSCC and, due to its purity from stromal contaminants and high ALDH1A1 expression, it represents an ideal in vitro model to understand HNSCC biology and test therapeutic applications. The observation that no other patients’ cells were able to grow anchorage-independent demonstrates how rare cells with these properties are in most HNSCC tumor samples and the strong variation among distinct tumor samples in biological parameters. This indicates that our culture system maintains the individual characteristics of distinct patients’ tumors and thus might be ideal to investigate therapeutic strategies in personalized approaches. Primary cell cultures of several patients showed signs of exhaustion, represented by lower proliferation, CAF outgrowth, and/or increased cell death. However, most samples proliferated for six or more passages in cell culture, providing enough time to expand and freeze sufficient amounts of cells available, e.g., for drug testing or personalized therapy approaches. In contrast to permanent cell lines, which show unlimited growth beyond passage 20, our primary cells are cultured only for short periods, preserving each patient’s characteristics. Due to a success rate of two-thirds using our system, fresh primary cultures can be continually established.

In summary, we developed a HNSCC cell culture model that facilitates the efficient expansion of epithelial tumor cells directly from patient-derived tumor tissue, which might be implemented successfully for other tumor types as well, providing valuable primary HNSCC cultures for future studies.

## Figures and Tables

**Figure 1 cells-08-00555-f001:**
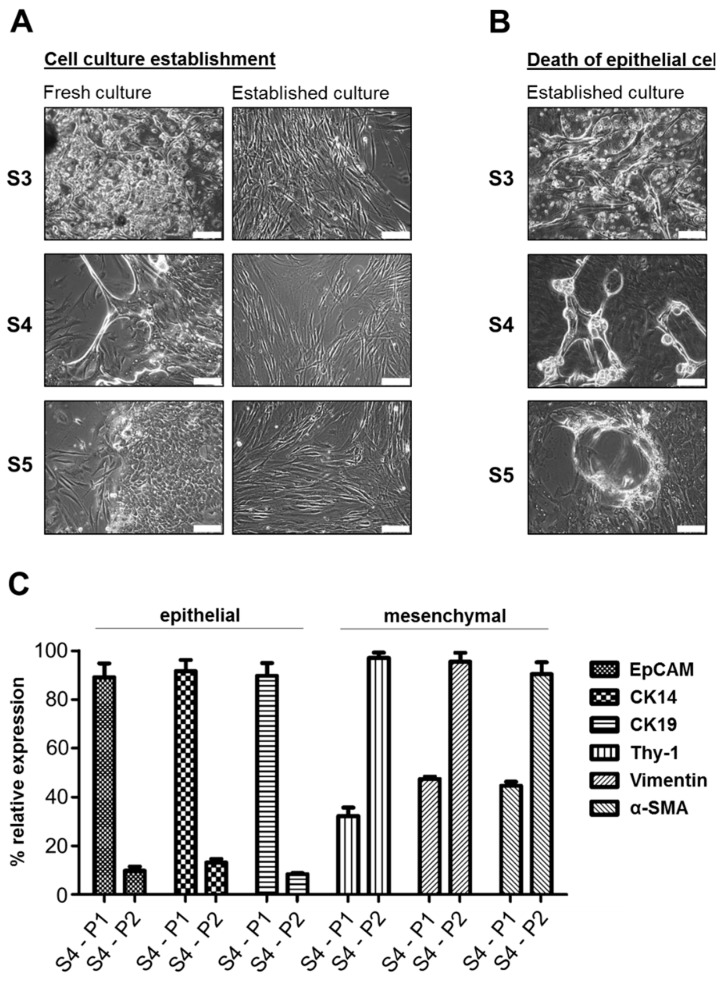
HNSCC cell culture establishment in culture medium containing 10% FBS. (**A**) Fresh patient-derived cultures of S3, S4, and S5 show mixed populations of epithelial and mesenchymal cells. By passaging, epithelial cells were gradually lost in the established cultures and expanding cells displayed fibroblast-like morphology; scale bars = 100 µm. (**B**) Epithelial cells developed scattered patches of cornifying cells; scale bars = 100 µm. (**C**) qRT-PCR analysis of passages 1 and 2 (P1 and P2) of S4 revealed decreasing expression of epithelial markers and an increase of mesenchymal markers.

**Figure 2 cells-08-00555-f002:**
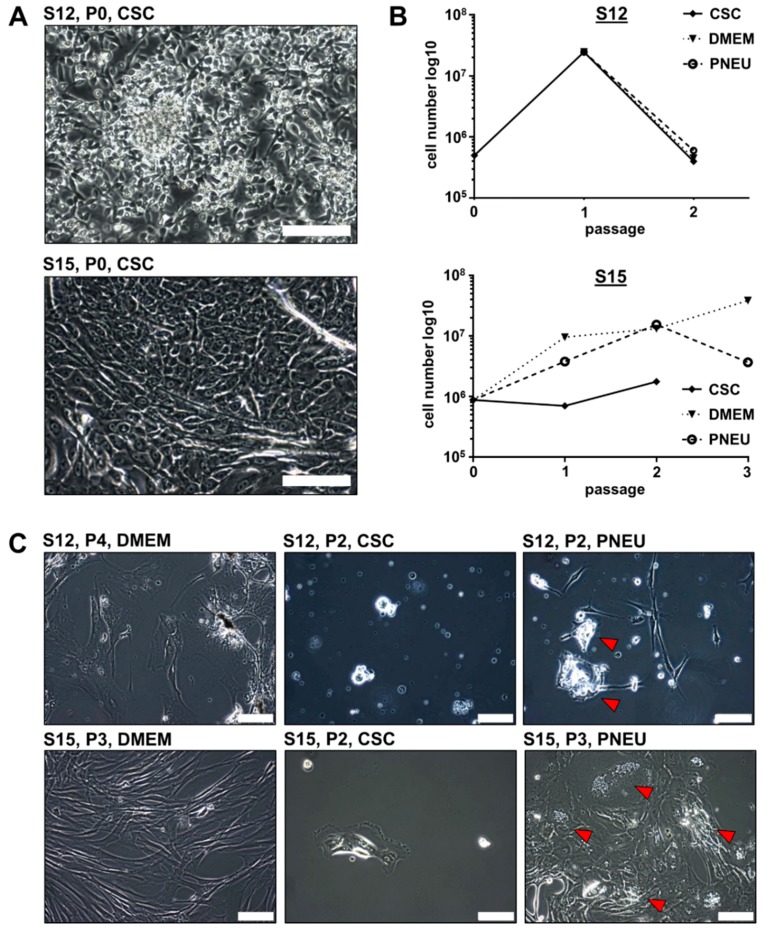
HNSCC cell culture establishment in serum-free CSC-medium. (**A**) Fresh primary cells of S12 and S15 display large epithelial cell compartment. (**B**) Growth curves show cell proliferation of S12 cells until passage 1 in CSC-medium followed by a decline in cell numbers after splitting into three different media. S15 cells expanded in DMEM and PNEU-medium, but showed poor proliferation in CSC-medium. (**C**) Upon initial passaging, cells were split into DMEM, CSC, and PNEU-medium, which resulted in fibroblast-like cell expansion in DMEM; no or neglectable expansion of epithelial cells in CSC-medium; epithelial cells expanded in PNEU-medium until passage 2 of S12 and S15. Then, PNEU cultures displayed growth of fibroblast-like cell types, while epithelial cells frequently acquired a phenotype reminiscent of cornification (arrows); scale bars = 100 µm.

**Figure 3 cells-08-00555-f003:**
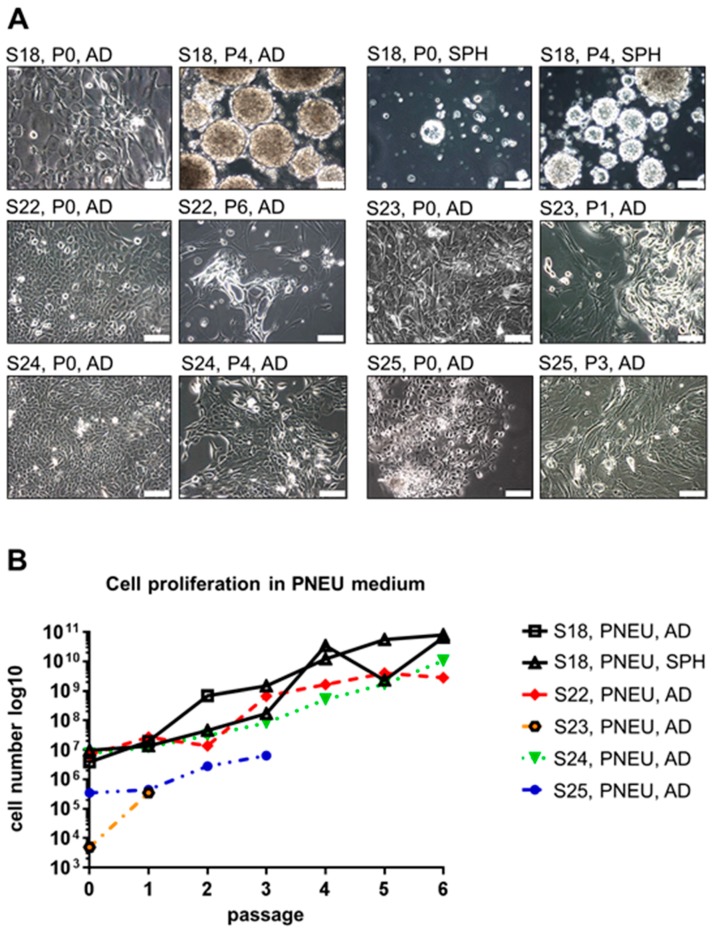
Primary HNSCC cell cultures in PNEU-medium. (**A**) Initial cultures in passage 0 (P0) and critical events observed in later passages: adherent S18 cells spontaneously transitioned into spheroids in passage 4; epithelial S22 cells expanded until passage 6 and then showed signs of reduced viability and were overgrown by fibroblast-like cells; epithelial cells derived from S23 expanded initially, but in passage 1 these cells exhibited death accompanied by proliferation of fibroblast-like cells; S24-derived cultures contained epithelial cells that expanded continuously until the experiment was terminated in passage 6; S25 cells with epithelial morphology expanded continuously until the experiment was terminated in passage 3; scale bars = 100 µm. (**B**) Growth curves of primary HNSCC cell cultures reveal varying proliferation rates; AD = adherent; SPH = spheroid.

**Figure 4 cells-08-00555-f004:**
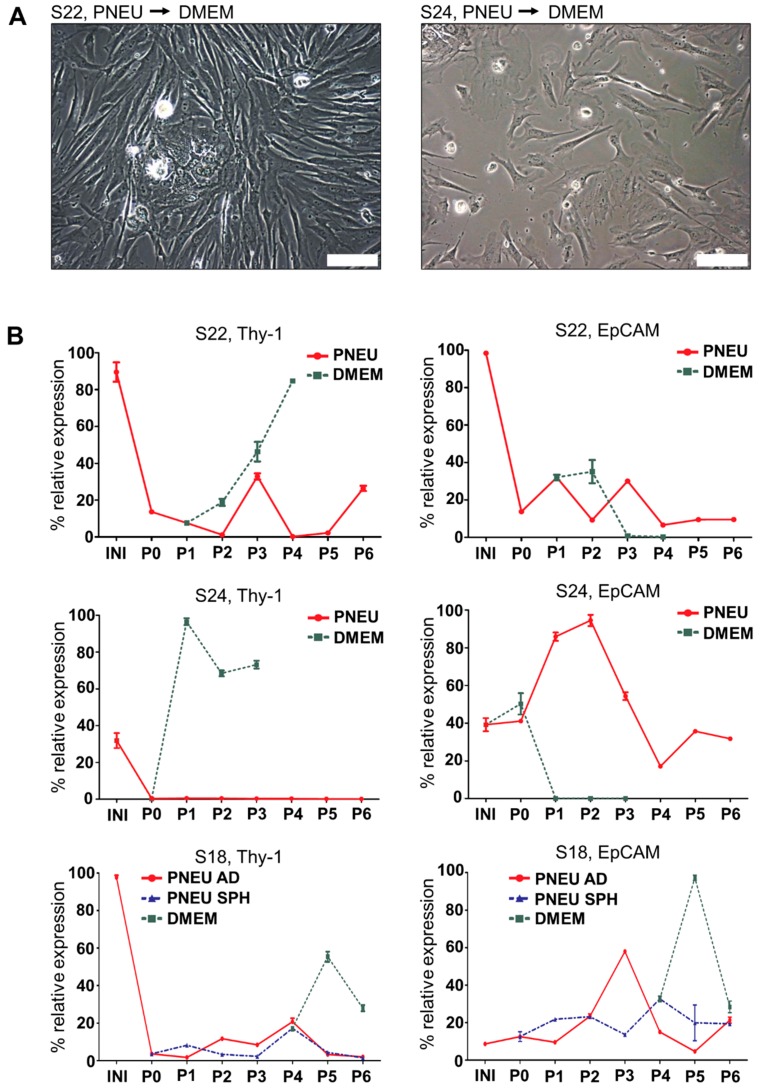
Serum treatment of primary HNSCC cell cultures. (**A**) In DMEM, fibroblast-like cell types replaced epithelial cells in adherent cultures of S22 and S24; scale bars = 100 µm. (**B**) The outgrowth of fibroblasts in DMEM-cultured S22 and S24 cells shown in (**A**) is accompanied by increased Thy-1 expression and loss of EpCAM expression over six culture passages. Adherent (PNEU AD) and spheroid (PNEU SPH) cells of S18 show constant expression of both markers and inconsistent changes in DMEM + 10% serum; INI = initial tumor tissue; P1 = passage 1.

**Figure 5 cells-08-00555-f005:**
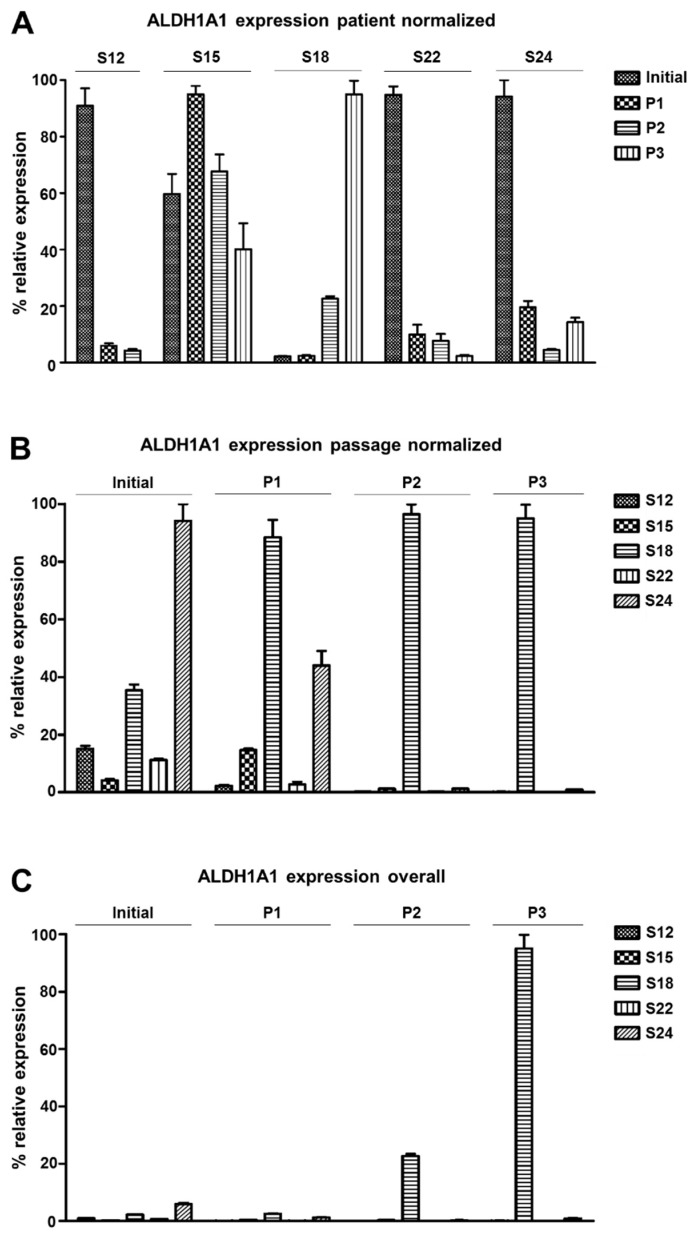
ALDH1A1 expression in primary HNSCC cell cultures in the initial tumor tissue and within the first three culture passages. (**A**) Relative expression (%) normalized by patient to compare ALDH1A1 levels relative to other cell entities of the same patient sample show higher ALDH1A1 levels in S18 cultures than in the initial S18 tumor. All other cultures showed lower or similar ALDH1A1 expression in comparison to the original tumor tissue. (**B**) Passage normalized data to compare ALDH1A1 levels between distinct patient samples within each passage and the initial tumors. Tumor sample S24 shows highest ALDH1A1 expression, whereas in cell culture cells of S18 overtop S24 cells form passage 1 on. (**C**) Data normalized to compare all samples of all passages and the initial tumor to each other reveals stark gradual enrichment of ALDH1A1 expressing cells in S18 cultures; P1 = passage 1; initial = original tumor tissue.

**Figure 6 cells-08-00555-f006:**
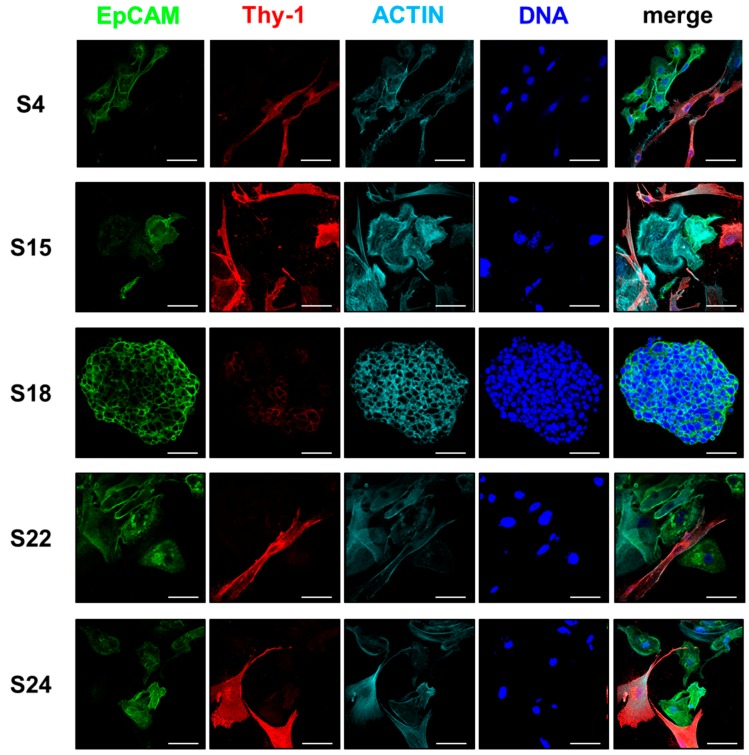
EpCAM and Thy-1 expression in mixed cell cultures composed of epithelial and mesenchymal populations visualized by indirect immunofluorescence. Cells with epithelial morphology stained highly positive for EpCAM, whereas cells with fibroblast-like morphology display high expression of Thy-1. Both markers are mutually exclusive in cell cultures derived from patients S4, S15, and S24. EpCAM positive cells of S18 and S22 showed weak Thy-1 cell membrane localization. Actin was stained using phalloidin to visualize cell morphology; DNA was stained using Hoechst33342; scale bars: 50 µm.

**Table 1 cells-08-00555-t001:** Specific markers for the distinction of primary epithelial and stromal cells derived from individual HNSCC tissue samples.

Markers	Epithelial Cells					Stroma Cells					Specific Marker
	S4	S15	S18	S22	S24	S4	S15	S18 *	S22	S24	
EpCAM	++	++	++	++	++	−	−	n.d.	−	−	epithelial
CK14	++	+	+	++	+	+	−	n.d.	−	−	No
CK19	++	++	++	++	++	−	−	n.d.	−	−	epithelial
Thy-1	−	−	+	−	−	++	++	n.d.	++	++	stromal
Vimentin	+	+	++	+	++	++	++	n.d.	++	++	No
α- SMA	+	+	+	+	+	++	+	n.d.	++	++	No

Marker analysis in primary cell cultures from S4, S15, S18, S22, and S24 by indirect immunofluorescence. − negative; + positive; ++ highly positive; n.d. = not determined; * no stroma culture was established from S18.

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
