# Peer review of "An Effective Primary Head and Neck Squamous Cell Carcinoma In Vitro Model"

_cells, 2019, doi:10.3390/cells8060555_

Round 1
Reviewer 1 Report
To make the results clearer, I would suggest a table showing what happened to which patient sample: Sample ID, tumor origin, p16 status, DMEM/CSC/PNEU...
The epithelial cells with vimentin and a-SMA expression caught my attention. In my opinion, functional tests for EMT or cell fusion are possible but would lead away from the main topic of this article. Because the paper is about establishment of primary HNSCC culture. Information about this procedure are very important for me. Additionally, the authors are right about the problems with the establishment of HPV positive cultures. Good to know that PNEU medium has the potential to cultivate HPV positive primary material.
It is possible to use ALDH1A1 as a marker for CSC. Of course, CSC's are a complex field and there are other markers, too. But unlike CD44 for example, ALDH1A1 is a selective marker that clearly represents a subpopulation. But it may does not show all CSC.
The ingredients of the PNEU medium are not known. STEMCELL is a trustworthy company. But details about the growth factors/inhibitors in the medium would advance the field very much. Additionally, the examination of the primary cultures for marker expression is very extensive.
Author Response
Response to reviewer #1:
To make the results clearer, I would suggest a table showing what happened to which patient sample: Sample ID, tumor origin, p16 status, DMEM/CSC/PNEU...
Answer: Yes, that is correct. We added “Supplementary Table S1” to this submission.
The epithelial cells with vimentin and a-SMA expression caught my attention. In my opinion, functional tests for EMT or cell fusion are possible but would lead away from the main topic of this article. Because the paper is about establishment of primary HNSCC culture. Information about this procedure are very important for me. Additionally, the authors are right about the problems with the establishment of HPV positive cultures. Good to know that PNEU medium has the potential to cultivate HPV positive primary material.
Answer: The putative fusion-cells are a very small subpopulation in our cultures. We noted their existence just to show that cells co-expressing EpCam and Thy-1 can be found in rare cases, but show irregular phenotypes. We agree that cell fusion or EMT assays would not advance the manuscript.
It is possible to use ALDH1A1 as a marker for CSC. Of course, CSC's are a complex field and there are other markers, too. But unlike CD44 for example, ALDH1A1 is a selective marker that clearly represents a subpopulation. But it may does not show all CSC.
Answer: This is right. ALDH1A1 is a good marker for CSCs in HNSCCs, but it might not cover all CSC-like cells. We used it to show that at least some CSC-like cells are contained in our cultures. CD44 has many isoforms which differ strongly in their function and significance between different tumor types (reviewed in PMID: 28986884). Thus, we decided that ALDH1A1 is a more reliable marker for our purpose.
The ingredients of the PNEU medium are not known. STEMCELL is a trustworthy company. But details about the growth factors/inhibitors in the medium would advance the field very much. Additionally, the examination of the primary cultures for marker expression is very extensive.
Answer: It will be important to determine the factors that support the epithelial growth in PNEU medium. However, STEMCELL Technologies is unlikely to help us with that.

Reviewer 2 Report
In this manuscript, the authors established several head and neck squamous cell carcinoma (HNSCC) cell lines that have been maintaining epithelial properties without the effectiveness of fibroblast-like cells.
I am very interested in their established HNSCC cell line because these cell lines grow keeping epithelial properties in medium without FBS. In particular, the S18 cell line may have the capacity of cancer stem cells. Therefore, I think this manuscript will be of interest to most HNSCC researchers in the world.
However, I have several concerns regarding this manuscript.
1. In Table 1, I think that the authors are trying to show whether the primary cells of each cell line have epithelial markers and/or mesenchymal markers. However, it is not clear why it is divided into epithelial cells and stroma cells. Also, how the authors detected the markers is unclear - was it using mRNA levels? I believe that the authors should provide further detail in the figure legend section.
2. In the discussion section, the authors should consider showing data in figures which are referred to in the text. For example, on page 18, lines 391 to 393, the authors wrote “however, in our study CAF-like cells outcompeted epithelial tumor cells in serum-containing medium and our data show that it compromises the proliferation and survival of epithelial tumor cells.” Please show the figure number that the authors want to indicate at the end of the sentence. Not only this, but there are other sentences in the discussion section with no indication of the relevant figure. I think that it is better to add figure numbers to all instances.
3. There are no S18 cell line data for changes via the passage of mesenchymal markers and epithelial markers, like in Fig.S3. I believe that the authors should show the data because I think this S18 cell line is interesting for most researchers.
Author Response
Response to reviewer #2:
In this manuscript, the authors established several head and neck squamous cell carcinoma (HNSCC) cell lines that have been maintaining epithelial properties without the effectiveness of fibroblast-like cells.
I am very interested in their established HNSCC cell line because these cell lines grow keeping epithelial properties in medium without FBS. In particular, the S18 cell line may have the capacity of cancer stem cells. Therefore, I think this manuscript will be of interest to most HNSCC researchers in the world.
However, I have several concerns regarding this manuscript.
1. In Table 1, I think that the authors are trying to show whether the primary cells of each cell line have epithelial markers and/or mesenchymal markers. However, it is not clear why it is divided into epithelial cells and stroma cells. Also, how the authors detected the markers is unclear - was it using mRNA levels? I believe that the authors should provide further detail in the figure legend section.
Answer: Table 1 shows the expression of epithelial and mesenchymal markers in epithelial cells and stromal cells isolated from the same tumor tissue sample. It is supposed to visualize which markers are specific for which cellular compartment. Confocal images of indirect immunofluorescence stained cells were used to create the table. We changed the title and the caption to clarify that.
Page 18, changed Table 1:
Table 1: Specific markers for the distinction of primary epithelial and stromal cells derived from individual HNSCC tissue samples.
2. In the discussion section, the authors should consider showing data in figures which are referred to in the text. For example, on page 18, lines 391 to 393, the authors wrote “however, in our study CAF-like cells outcompeted epithelial tumor cells in serum-containing medium and our data show that it compromises the proliferation and survival of epithelial tumor cells.” Please show the figure number that the authors want to indicate at the end of the sentence. Not only this, but there are other sentences in the discussion section with no indication of the relevant figure. I think that it is better to add figure numbers to all instances.
Answer: Figures present data which are referred to in the results section. To refer to figures in the discussion section would be unusual. Additionally, in the discussion section we do not refer to data in specific figures, but rather to the overall conclusions which may derive from several figures. We can add figure numbers, if the editor agrees to that.
3. There are no S18 cell line data for changes via the passage of mesenchymal markers and epithelial markers, like in Fig.S3. I believe that the authors should show the data because I think this S18 cell line is interesting for most researchers.
Answer: Yes, that makes sense. We added the data to Figure 4 on page 13:
Figure 4: Serum-treatment of primary HNSCC cell cultures. (A) In DMEM, fibroblast-like cell types replaced epithelial cells in adherent cultures of S22 and S24; scale bars = 100 µm. (B) The outgrowth of fibroblasts in DMEM-cultured S22 and S24 cells shown in (A) is accompanied by increased Thy-1 expression and loss of EpCAM expression over six culture passages. Adherent (PNEU AD) and spheroid (PNEU SPH) cells of S18 show constant expression of both markers and inconsistent changes in DMEM + 10% serum; INI = initial tumor tissue; P1 = passage 1.

Reviewer 3 Report
The authors described an effective primary culture method for HNSCC. This manuscript is interesting and well written; however, there are some issues to resolve in this manuscript.
1. In the figure 1C, 4B and S3, what did the relative expression mean? What is the control (that means what the “100%” is)?
2. In the figure 2B and 3B, number of the cells was shown. How and when did the authors count the cells? Which plates or dishes were used? In the figure 3B, the number of S18P6 cells was more than 10 billion. Was a single culture plate used?
3. Please show the title of the vertical axis in the figure 5.
4. As the authors described in the discussion section, some kind of primary cancer cells could be grown in the traditional media with serum, but the HNSCC cells can’t. The authors should discuss more why the primary cancer cells from HNSCC tissue require the specific media.
Author Response
Response to reviewer #3
The authors described an effective primary culture method for HNSCC. This manuscript is interesting and well written; however, there are some issues to resolve in this manuscript.
1. In the figure 1C, 4B and S3, what did the relative expression mean? What is the control (that means what the “100%” is)?
Answer: The relative expression value is measured in relation to the Glyceraldehyde 3-phosphate dehydrogenase (GAPDH) expression. The 100% is the highest expression value in the analysis and all other values are normalized to that value. This is why we have indicated the parameter, by which the data has been normalized for the figure 5.
For clarification, in the material and methods section 2.5 we have changed the manuscript:
Page 7, paragraph 1:
“Glyceraldehyde 3-phosphate dehydrogenase (GAPDH) expression was utilized for relativization of cycle threshold values. GraphPad Prism (GraphPad Software) was used for graphics and statistical analysis. The highest expression value in the analysis was normalized to 100%.”
2. In the figure 2B and 3B, number of the cells was shown. How and when did the authors count the cells? Which plates or dishes were used? In the figure 3B, the number of S18P6 cells was more than 10 billion. Was a single culture plate used?
Answer: The numbers are theoretical and calculated under the assumption that all bottles would have proliferated the same way as the cells in the examined bottle. The number of cells counted was multiplied by the split rate. Thus, if 1 million cells were counted and the cells were previously passaged 4 times 1:4, the theoretical cell number was 4 x 4 x 4 x 4 x 1 million = 256 million cells. This is the standard procedure to monitor exponential cell growth.
In the material and methods section 2.3 we have added:
Page 5, paragraph 2:
“Cell numbers were calculated using a Neubauer Chamber. To monitor exponential growth the counted cell number was multiplied by the split rate.”
3. Please show the title of the vertical axis in the figure 5.
Answer: We have added the vertical axis title “% relative expression” and added in the legend that “%” relates to the relative expression.
Page 15, updated Figure 5:
Figure 5: ALDH1A1 expression in primary HNSCC cell cultures in the initial tumor tissue and within the first three culture passages. (A) Relative expression (%) normalized by patient to compare ALDH1A1 levels relative to other cell entities of the same patient sample show higher ALDH1A1 levels in S18 cultures than in the initial S18 tumor. All other cultures showed lower or similar ALDH1A1 expression in comparison to the original tumor tissue. (B) Passage normalized data to compare ALDH1A1 levels between distinct patient samples within each passage and the initial tumors. Tumor sample S24 shows highest ALDH1A1 expression, whereas in cell culture cells of S18 overtop S24 cells form passage 1 on. (C) Data normalized to compare all samples of all passages and the initial tumor to each other reveals stark gradual enrichment of ALDH1A1 expressing cells in S18 cultures; P1 = passage 1; initial = original tumor tissue.
4. As the authors described in the discussion section, some kind of primary cancer cells could be grown in the traditional media with serum, but the HNSCC cells can’t. The authors should discuss more why the primary cancer cells from HNSCC tissue require the specific media.
Answer: Our data clearly indicate that primary HNSCC cells have specific requirement regarding the culture medium. We believe that the reason is that in classical medium HNSCC cells are simply overgrown by stromal fibroblasts and that unknown factors in classical medium might trigger a cornification-like differentiation.
We added the following explanation on page 19, paragraph 2:
“The reason, why primary HNSCC cells require specific culture medium conditions, remains unknown. However, PNEU-medium appears to protect epithelial cells and facilitates their proliferation sufficiently, so that they are not overgrown by fibroblasts. Furthermore, specific factors in classical media might trigger the cornification observed here.”

Round 2
Reviewer 2 Report
The authors have accepted my suggestions and have answered my questions. Thank you.